# Condensing Heterogeneous Datasets without Domain Labels

## Abstract

Dataset Condensation (DC) is a powerful technique for reducing large-scale training costs, but its effectiveness is largely confined to homogeneous data. When confronted with heterogeneous datasets from multiple sources, existing DC methods falter, often collapsing toward dominant visual styles and discarding crucial domain-specific information. To address this critical limitation, we propose Condensing Heterogeneous Datasets without Domain Labels (CHDDL), a novel framework that embeds rich domain diversity directly into synthetic images. CHDDL achieves this through a domain-aware module that employs learnable spatial masks, guided by a lightweight and entirely unsupervised FFT-based pseudo-labeling scheme. Crucially, our approach operates without requiring explicit domain labels and preserves the original Images Per Class (IPC) budget, making it a practical, plug-and-play enhancement for existing DC methods. Extensive experiments demonstrate that CHDDL consistently outperforms strong baselines across single-domain, multi-domain, and cross-architecture generalization settings, highlighting its potential as a key component for robust dataset condensation in realistic, multi-source environments.

## 1 Introduction

Over the past decade, deep learning models have grown substantially in capacity, achieving remarkable progress in diverse tasks across vision, language, and multi-modal domains. This performance growth has been tightly coupled with the dataset size. To meet this demand, data collection has shifted from manual curation to automated web crawling, yielding datasets that are not only large but also highly heterogeneous, often consisting of samples drawn from various domains with drastically different visual characteristics. While such diversity benefits model robustness, it also raises new challenges for training efficiency at scale.

Dataset Condensation (DC) has emerged as a promising direction to reduce this training cost by synthesizing a small set of highly informative samples. First introduced by Wang et al. (Wang et al., 2018), DC replaces the original training data with a compact synthetic dataset, optimized to preserve the training dynamics of real data. Subsequent methods have improved upon this core idea using techniques like gradient matching (Zhao et al., 2021), distribution alignment (Zhao & Bilen, 2023), and trajectory matching (Cazenavette et al., 2022), showing strong results on curated benchmarks.

However, the success of these methods rests on an implicit assumption: that the training data is relatively homogeneous. This assumption breaks down in many realistic scenarios where datasets are aggregated from multiple unlabeled sources. When conventional DC is applied to such mixed-domain data, the synthetic images often converge toward the features of the most dominant or visually simple domains. This leads to a loss of diversity and critical information from minority domains, ultimately degrading the condensed set's overall performance and generalization capability.

This failure is not merely theoretical. As demonstrated in Figure 1, a significant performance gap exists between condensing on a single, isolated domain versus a full, multi-domain dataset. This gap provides clear evidence that existing methods struggle to preserve domain-specific characteristics in a heterogeneous setting. A seemingly straightforward solution is to condense each domain separately. However, this approach is fundamentally impractical in real-world settings for two key reasons:

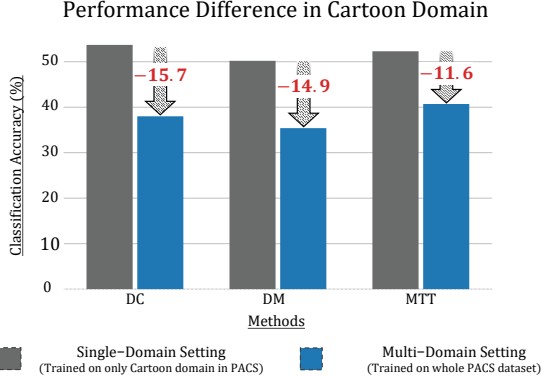

Figure 1: Evidence of Performance Degradation for Conventional DC Methods in a Multi-Domain Setting. The experiment compares two scenarios on the PACS dataset (Li et al., 2017) under a 10 Image Per Class (IPC), with all models evaluated on the 'Cartoon' domain. In the Single-Domain Condensation setting (Gray Bars), data is condensed using only images from the target 'Cartoon' domain, representing an ideal but unrealistic case with oracle label access. The Multi-Domain Condensation setting (Blue Bars) involves condensing the full, domain unlabeled PACS dataset, reflecting a practical scenario. The consistent and significant performance drop up to 15.7% across all methods demonstrates that they fail to preserve crucial domain-specific features when faced with heterogeneous data, motivating the need for our proposed MDDC task.

1) Lack of Labels: The very premise of the problem we address is that explicit domain labels are unavailable for separating the data. 2) Budget Inflation: Even if labels were available, creating separate condensed sets for each domain would multiply the synthetic dataset's size by the number of domains, violating the core efficiency goal of DC.

The limitations of existing methods and the impracticality of simple alternatives necessitate a new research direction. We therefore formally define the task of Multi-Domain Dataset Condensation (MDDC). The goal of MDDC is to synthesize a single, compact dataset that effectively generalizes across multiple domains under the challenging constraints of (1) having no access to explicit domain labels and (2) adhering to a fixed Images Per Class (IPC) budget. To tackle the MDDC challenge, we propose Condensing Heterogeneous Datasets without Domain Labels (CHDDL), a novel, plug-and-play framework. CHDDL introduces a domain-aware module, which embeds diverse domain features into each synthetic image via learnable spatial masks. To guide this process without supervision, we leverage a lightweight FFT-guided pseudo-domain labeling scheme that captures latent domain characteristics from low-frequency

amplitude statistics. Because the domain-aware module is active only during the one-time condensation process, CHDDL enhances the final synthetic data without increasing the IPC budget or adding overhead to downstream model training.

The main contributions of this work are as follows:

- We are the first to identify the performance degradation of DC methods in heterogeneous settings and formally define the Multi-Domain Dataset Condensation (MDDC) task with its practical constraints.

- We propose CHDDL, the first effective solution for the MDDC task, which uniquely combines unsupervised FFT-based pseudo-labeling with learnable spatial masks to embed domain diversity without requiring domain labels or increasing the dataset size (i.e., IPC).

- We demonstrate through extensive experiments on five benchmark datasets that applying CHDDL to various baseline methods consistently improves in-domain, out-of-domain, and cross-architecture performance, showcasing its effectiveness and wide applicability.

## 2 RELATED WORKS

### 2.1 DATASET CONDENSATION

As machine learning models have become larger and more complex, the amount of data required for training those models has also grown significantly. In this context, the emergence of massive datasets has greatly increased the burden on computational resources and training time, creating a bottleneck in model development. Dataset distillation (Wang et al., 2018) is a formulation proposed to address this issue by compressing a large dataset into a much smaller synthetic dataset while still maintaining the essential data characteristics of the original dataset for training deep learning models. This approach drastically reduces training time and computational costs, allowing models

trained on the condensed dataset to achieve performance comparable to those trained on the original, large-scale datasets. Among various strategies in dataset condensation, including gradient matching methods (Wang et al., 2018; Zhao et al., 2021; Kim et al., 2022), approaches based on distribution matching (Zhao & Bilen, 2023; Zhao et al., 2023), trajectory matching (Cazenavette et al., 2022; Guo et al., 2024), decoupling method (Yin et al., 2023), and generative-model-based approaches leveraging GANs or diffusion models (Cazenavette et al., 2023; Su et al., 2024), we focus on gradient matching, distribution matching, and trajectory matching.

Dataset distillation methods based on **gradient matching** aim to match the gradients of a neural network that are calculated for a loss function over a synthetic dataset and the original dataset for the purpose of dataset condensation. DC (Zhao et al., 2021) first formulated the dataset distillation as a minimization problem between gradients that are calculated from an original dataset and a condensed dataset. IDC (Kim et al., 2022) improved data condensation by efficiently parameterizing synthetic data to preserve essential characteristics with a smaller dataset. Zhang et al. (Zhang et al., 2023) accelerated the distillation process by utilizing models in the early stages of training. Dataset distillation methods based on **distribution matching** were proposed to overcome the limitations of gradient matching methods, which require complex optimization and high computational costs. DM (Zhao & Bilen, 2023) introduced a method that aligns the distribution of the original and synthetic datasets in embedding space, significantly improving the efficiency of dataset distillation. IDM (Zhao et al., 2023) enhanced distribution matching by addressing class imbalance and embedding issues. Recently, **trajectory matching** based dataset condensation methods has been actively researched. MTT (Cazenavette et al., 2022) developed a method to create condensed datasets by mimicking the training trajectories of models trained on the original dataset. By aligning the synthetic dataset's training path with that of the original data, they significantly improved the efficiency of dataset distillation. FTD (Du et al., 2023) improved trajectory matching by addressing the accumulated trajectory error, which often led to discrepancies between training and evaluation performance. DATM (Guo et al., 2024) addressed limitations in prior dataset distillation methods by introducing difficulty-aligned trajectory matching.

### 2.2 DOMAIN-AWARE LEARNING APPROACHES

Research in domain-aware learning is crucial to addressing performance degradation caused by discrepancies between different domains. Machine learning models tend to perform optimally when the distribution of training data matches that of test data. However, in real-world applications, data is often collected across various domains with distinct distributions. These domain shifts can significantly impact a model's generalization performance; without addressing these differences, models may only be effective in limited, specific environments. Two prominent approaches to mitigate this issue are domain adaptation and domain generalization. Domain adaptation (Ganin et al., 2016; Long et al., 2018; Zhang et al., 2019; Zhu et al., 2023) focuses on improving the model's performance on a target domain by leveraging knowledge from a source domain where training data is available. This typically involves techniques that reduce distributional differences between the source and target domains or map features from both domains onto a common representation. In contrast, domain generalization (Li et al., 2018; Zhou et al., 2021; Cha et al., 2021; Yao et al., 2022; Choi et al., 2023) aims to build a model that can generalize to new, unseen domains without direct access to their data. Domain generalization methods utilize multiple source domains to create a robust model that would perform equally well in various unseen domains.

Our plug-and-play method for multi-domain dataset condensation is related to previous domain-aware learning methods as it differentiates domains within the training dataset and considers possible domain shifts. To the best of our knowledge, CHDDL (Condensing Heterogeneous Datasets without Domain Labels) is the first work to incorporate domain-awareness into dataset condensation, bridging a previously unexplored gap between dataset condensation and multi-domain dataset.

## 3 METHOD

Given a dataset $\mathcal{D}_{\text{real}} = \{x_n, y_n\}_{n=1}^{N}$ where $y_n \in \{0, \cdots, C-1\}$, single-domain dataset condensation aims to synthesize a much smaller synthetic dataset $\mathcal{D}_{\text{syn}} = \{\tilde{x}_m, \tilde{y}_m\}_{m=1}^{M}$ where $M \ll N$ such that $\mathcal{D}_{\text{syn}}$ has the same or similar power as $\mathcal{D}_{\text{real}}$ in terms of model training.

In Multi-Domain Dataset Condensation (MDDC), it takes a step further and encodes domain variability within the $\mathcal{D}_{\text{syn}}$ without explicit domain labels while preserving the class-discriminative features.

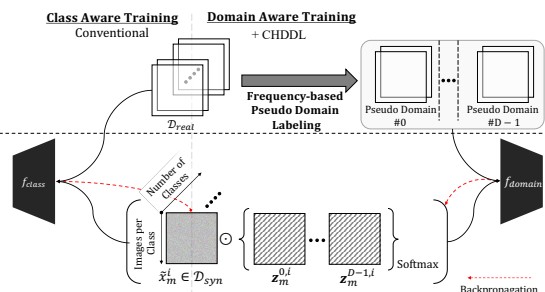

### 3.1 FREQUENCY-BASED PSEUDO DOMAIN LABELING

In many curated multi-domain benchmarks (e.g., PACS, Office-Home), explicit domain labels are available as domain differences are mostly distinguishable. However, for unconstrained web data or large mixed datasets, explicit domain labels are mostly unavailable, primarily because the goal of the dataset is not for classifying the domains but also because the distinction for each domain is vague or overlapping. We first define

Figure 2: Overview of Condensing Heterogeneous Datasets without Domain Labels (CHDDL). Our method augments class-aware training (left) with domain-aware learning (right) by embedding pseudo-domain features via learnable spatial masks. The masks are jointly optimized with synthetic images without increasing IPC.

*domain* as variation not attuned with task-relevant information, in this case class-discriminative features, and leverage Fast Fourier Transformation (FFT) to extract domain-specific information for each image in $\mathcal{D}_{\text{real}}$ as theoretically supported and applied in prior domain adaptation and domain generalization (Xu et al., 2021; Yang & Soatto, 2020). For every real image $x_n$ the discrete 2-D Fourier transform as follows

$$\mathcal{F}(x_n)[u,v] = \sum_{h=0}^{H-1}\sum_{w=0}^{W-1} x_n[h,w]\, e^{-j2\pi(uh/H+vw/W)}, \tag{1}$$

which is computed per color channel. Through shifting, the center of the amplitude becomes the low-frequency region, which prior domain adaptation and domain generalization methods leveraged for domain-specific information. Likewise, we crop the central region with a cropping ratio $\beta$ and get the mean of the amplitude, $\mu_n$, as follows:

$$\mu_n = \frac{1}{3\beta^2 HW}(Crop_\beta\{|\mathcal{F}(x_n)|_{\text{shifted}}\} \in \mathbb{R}^{\beta H \times \beta W \times 3}). \tag{2}$$

Sorting $\{\mu_n\}_{n=1}^N$ in ascending or descending order and slicing it into $D$ equal parts assigns pseudo-domain labels

$$d_n = \left\lfloor \frac{\text{ranking}_{(\mu_n)-1}}{N/D} \right\rfloor,$$
$$d_n \in \{0,\ldots,D-1\}, \quad \text{ranking}(\mu_n) \in \{1,\ldots,N\}.$$

### 3.2 DOMAIN-AWARE MODULE

Each synthetic image $\tilde{x}_m \in \mathbb{R}^{H \times W \times 3}$ is paired with a learnable domain mask $\mathbf{z}_m^{d,i} \in \mathbb{R}^{H \times W \times 3}, d = \{0,\ldots,D-1\}$, where $D$ is the number of pseudo domains and $i$ denotes the current iteration. We initialize all elements in the domain mask with 0.01 at $i=0$. To prevent a single domain dominating the synthetic image, we leverage a per-pixel temperated softmax to generate relative importance of each domain to each synthetic image, as well as to balance the domain importance among the $\mathbf{z}_m^{d,i}$ as follows:

$$\alpha_m^{d,i} = \frac{\exp(\mathbf{z}_m^{d,i}/\tau)}{\sum_{d'=0}^{D-1}\exp(\mathbf{z}_m^{d',i}/\tau)}, \tag{3}$$

where $\alpha_m^{d,i} \in \mathbb{R}^{H \times W \times 3}$ and $\tau$ is the temperature factor in the softmax function. Through $\alpha_m^{d,i}$, a synthetic image saliency map with domain $d$ at iteration $i$ is obtained as

$$\tilde{x}_m^{d,i} = \tilde{x}_m^i \odot \alpha_m^{d,i}, \tag{4}$$

where $\odot$ is element-wise multiplication, and this satisfies the exact reconstruction identity as

$$\tilde{x}_m^i = \sum_{d=0}^{D-1} \tilde{x}_m^{d,i}, \tag{5}$$

since $\sum_{d=0}^{D-1} \alpha_m^{d,i} = 1$ as it is output of softmax function.

Through a domain-aware module, several domains can coexist in disjoint spatial regions without information loss and without increasing number of synthetic images. Note that $\mathbf{z}_m^{d,i}$ is trained along with $\tilde{x}_m^i$ during training. As both $\mathbf{z}_m^{d,i}$ and $\tilde{x}_m^i$ are trained, $\tilde{x}_m^{d,i}$ are updated by Equation 4.

### 3.3 TRAINING OBJECTIVE

Being a plug-and-play module, we leverage the same loss function from the prior base models $\mathcal{L}_{base}$ and the loss for the class becomes

$$\mathcal{L}_{\text{cls}} = \mathcal{L}_{base}(\Theta; \mathcal{D}_{\text{real}}, \mathcal{D}_{\text{syn}}), \tag{6}$$

where $\Theta$ is the parameters needed for the loss computation. Accordingly, we define the domain loss as

$$\mathcal{L}_{\text{dom}} = \mathcal{L}_{base}(\Theta'; \mathcal{D}_{\text{real}}, \mathcal{D}_{\text{syn}}^{\text{dom}}). \tag{7}$$

Here, parameters for the domain loss are denoted as $\Theta'$. The $\mathcal{D}_{\text{syn}}^{\text{dom}} = \left\{ \tilde{\mathbf{x}}_m^d, \tilde{y}_m \right\}_{1 \leq m \leq M, 0 \leq d < D}$ where $\tilde{x}_m^d = \tilde{x}_m \odot \alpha_m^d$, is used solely to supervise the domain-aware module during the condensation phase. **Notably, the domain masks $\mathbf{z}_m^{d,i}$ used to compute $\alpha_m^d$ are discarded after condensation.** As a result, only $\tilde{x}_m \in \mathcal{D}_{syn}$ is used during downstream training, ensuring that the number of synthetic images remains unchanged and the Images per Class is preserved.

The architecture for domain loss is identical to class loss, but the parameters are initialized differently. Also, note that the $\mathcal{D}_{real}$ for class and domain loss is the same while the batch configuration differs. For the class loss, the batch is grouped by the class label following the prior methods, while it is grouped by the pseudo-domain label for the domain loss. To sum up, the final loss becomes

$$\mathcal{L}_{\text{total}} = \mathcal{L}_{\text{cls}} + \lambda \mathcal{L}_{\text{dom}}, \tag{8}$$

where $\lambda$ is the weighting factor. $\mathcal{L}_{\text{cls}}$ provides gradients for updating $\tilde{x}_m^i$ and $\mathcal{L}_{\text{dom}}$ for updating both $\tilde{x}_m^i$ and $\mathbf{z}_m^{d,i}$. Parameters $\Theta$ and $\Theta'$ are randomly initialized or frozen after being trained on real data, depending on the prior method (Details are given in the Appendix). The overall pipeline is illustrated in Figure 2.

## 4 EXPERIMENTS

### 4.1 DATASET

We evaluate our method, CHDDL, on $32 \times 32$ CIFAR-10 and CIFAR-100 (Krizhevsky et al., 2009), and $64 \times 64$ Tiny ImageNet (Le & Yang, 2015), the three most commonly used datasets in the field of dataset condensation. The experiment setting with these datasets is **single-domain setting**. Additionally, we employ $64 \times 64$ PACS (Li et al., 2017), VLCS (Fang et al., 2013), and Office-Home (Venkateswara et al., 2017) datasets that are commonly used in the field of domain adaptation (DA) and domain generalization (DG). These multi-domain datasets have four distinct domains and are leveraged not only to validate the effectiveness of CHDDL in **multi-domain setting** but also to better analyze the differences between single- and multi-domain dataset settings. We note that the provided domain labels are not leveraged unless explicitly stated in the experiment setting.

### 4.2 IMPLEMENTATION DETAILS

We implement CHDDL on three pioneering prior methods, DC (Zhao et al., 2021), DM (Zhao & Bilen, 2023), and MTT (Cazenavette et al., 2022), in gradient matching, distribution matching, and trajectory matching dataset condensation, respectively. For a fair comparison, we follow the conventional experiment settings employing ConvNet architecture (Gidaris & Komodakis, 2018) while varying the depth of the network depending on the image size of the $D_{real}$. More specifically, three-depth ConvNet is utilized for all experiments with the CIFAR-10 and CIFAR-100 datasets, while all the other datasets leverage four-depth ConvNet. All of the hyperparameters introduced in each prior method are set identically, and the learning rate for the DG datasets is set equal to the Tiny ImageNet setting. We note that $D_{syn}$ is initialized with **Gaussian noise** in all of our experiments rather than initializing with **real** image from $D_{real}$ as some prior works do. However,

Table 1: Results with and without CHDDL on the prior methods on the **single-domain** setting. "T.Image." denotes Tiny ImageNet dataset. All results are the average of 10 runs and reported as mean $\pm$ standard deviation.

| Dataset | CIFAR-10 | | | CIFAR-100 | | | T.Image. |
|---|---|---|---|---|---|---|---|
| Img/Cls | 1 | 10 | 50 | 1 | 10 | 50 | 1 |
| Ratio (%) | 0.02 | 0.2 | 1 | 0.2 | 2 | 10 | 0.2 |
| Random | $12.5_{\pm 0.8}$ | $25.1\pm1.4$ | $42.5\pm0.5$ | $3.7\pm0.2$ | $13.9\pm0.3$ | $29.0\pm0.3$ | $1.3\pm0.1$ |
| DC | $27.4\pm0.2$ | $43.3\pm0.3$ | $53.0\pm0.3$ | $12.2\pm0.3$ | $24.8\pm0.3$ | - | - |
| **DC + CHDDL** | $\mathbf{29.0}\pm0.5$ | $\mathbf{45.4}\pm0.3$ | $\mathbf{54.5}\pm0.2$ | $\mathbf{13.0}\pm0.2$ | $\mathbf{25.8}\pm0.1$ | - | - |
| DM | $24.7\pm0.3$ | $47.4\pm0.4$ | $58.2\pm0.1$ | $10.9\pm0.2$ | $29.2\pm0.2$ | $36.5\pm0.2$ | $3.7\pm0.1$ |
| **DM + CHDDL** | $\mathbf{27.1}\pm0.3$ | $\mathbf{49.8}\pm0.5$ | $\mathbf{59.5}\pm0.2$ | $\mathbf{11.8}\pm0.2$ | $\mathbf{30.0}\pm0.1$ | $\mathbf{37.3}\pm0.2$ | $\mathbf{4.2}\pm0.1$ |
| MTT | $41.9\pm0.4$ | $50.7\pm0.8$ | - | $15.8\pm0.3$ | $35.3\pm0.2$ | - | $4.8\pm0.3$ |
| **MTT + CHDDL** | $\mathbf{46.8}\pm0.4$ | $\mathbf{57.9}\pm0.4$ | - | $\mathbf{24.0}\pm0.3$ | $\mathbf{35.9}\pm0.2$ | - | $\mathbf{5.7}\pm0.2$ |
| Whole Dataset | $84.8\pm0.1$ | | | $56.2\pm0.3$ | | | $37.6\pm0.4$ |

Table 2: Results with and without CHDDL on the prior methods on the **multi-domain** setting. All results are the average of 10 runs and reported as mean $\pm$ standard deviation.

| Dataset | PACS | | VLCS | | Office-Home | |
|---|---|---|---|---|---|---|
| Img/Cls | 1 | 10 | 1 | 10 | 1 | 10 |
| Ratio (%) | 0.08 | 0.8 | 0.07 | 0.7 | 0.46 | 4.6 |
| Random | $18.1\pm2.6$ | $33.0\pm0.8$ | $17.3\pm2.2$ | $27.0\pm1.6$ | $3.9\pm0.3$ | $12.9\pm0.6$ |
| DC | $35.3\pm0.6$ | $46.1\pm0.7$ | $29.6\pm0.9$ | $39.0\pm0.6$ | $11.0\pm0.3$ | - |
| **DC + CHDDL** | $\mathbf{38.8}\pm0.7$ | $\mathbf{48.3}\pm0.5$ | $\mathbf{34.8}\pm1.0$ | $\mathbf{42.7}\pm0.5$ | $\mathbf{12.4}\pm0.4$ | - |
| DM | $28.7\pm0.5$ | $46.7\pm0.5$ | $29.1\pm1.7$ | $42.0\pm0.3$ | $9.0\pm0.3$ | $25.5\pm0.3$ |
| **DM + CHDDL** | $\mathbf{34.7}\pm1.1$ | $\mathbf{50.9}\pm0.4$ | $\mathbf{36.7}\pm1.1$ | $\mathbf{44.4}\pm0.3$ | $\mathbf{10.4}\pm0.3$ | $\mathbf{27.2}\pm0.3$ |
| MTT | $39.7\pm0.6$ | $45.9\pm0.8$ | $28.5\pm2.1$ | - | $13.8\pm0.2$ | - |
| **MTT + CHDDL** | $\mathbf{46.6}\pm0.9$ | $\mathbf{50.6}\pm0.6$ | $\mathbf{39.7}\pm1.8$ | - | $\mathbf{16.3}\pm0.2$ | - |
| Whole Dataset | $72.0\pm0.8$ | | $60.8\pm0.6$ | | $50.4\pm0.8$ | |

we demonstrate that the performance gap still persists even when initializing with real image in the Appendix. $\mathbf{z}_m^{d,i}$ is initialized with 0.01 and the temperature $\tau$ is set to 0.1 when applying softmax among the $D$ masks for all experiments. The domain embedding weight $\lambda$ is set to 0.1 for DC and DM and 0.01 for the MTT. All of the hyperparameter sweep experiments (e.g., $\mathbf{z}_m$ initial value, domain embedding weight $\lambda$, and temperature $\tau$ value) can be found in the Appendix. $D$ is set to 4 for all experiments. Finally, we follow DM (Zhao & Bilen, 2023) for the evaluation protocol for all the experiments, and the results presented in the tables are the average of 10 evaluation results.

## 4.3 RESULTS

**Main results** Table 1 demonstrates the performance on the single-domain setting with three commonly utilized benchmarks in dataset condensation by varying Image per Class (IPC) and prior dataset condensation methods employed along with CHDDL. Similarly, in Table 2, we showcase the performance on three commonly used benchmarks in DA and DG for the multi-domain setting. Note, we use all of the domains in the dataset for training and evaluating the DG datasets. The "-" in the tables denotes the experiment setting, which either 1) was not done in the original paper or 2) requires extensive computational resources beyond our limit. All reported experiments show performance improvements when leveraging CHDDL with the prior methods. All reported results are top-1 classification accuracies measured on the test split, using models trained solely on the condensed dataset. This consistent improvement confirms that domain-aware module effectively enriches condensed data with domain-specific structure while preserving class-discriminative information, the core objective in classification task. Despite the risk that embedding additional domain cues might interfere with class semantics, the observed gains demonstrate that domain-aware module successfully integrates domain context in a way that reinforces, rather than disrupts, the underlying class structure as intended. We visualize the final condensed synthetic data for CIFAR-

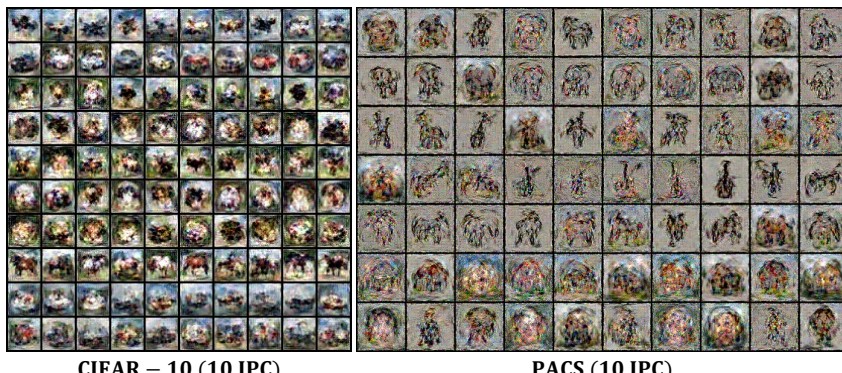

**CIFAR − 10** (**10 IPC**)            **PACS** (**10 IPC**)

Figure 3: Visualization of the final output in CIFAR-10 and PACS under 10 IPC setting. The shown images are condensed with DC+CHDDL. More outputs can be found in the supplementary material.

Table 3: Cross-architecture results with condensed CIFAR-10 data under 10 IPC with ConvNet on various architectures. All results are the average of 10 runs and reported as mean ± standard deviation.

| Method | ConvNet | AlexNet | VGG | ResNet18 | ViT-Tiny | ViT-Small |
|---|---|---|---|---|---|---|
| DC | 43.3±0.3 | 15.0±3.4 | 34.6±0.2 | 18.7±0.5 | 21.7±0.6 | 21.7±0.5 |
| **DC + CHDDL** | **45.4**±0.3 | **22.8**±1.2 | **35.9**±0.4 | **19.5**±0.6 | **22.4**±0.4 | **22.4**±0.4 |
| DM | 47.4±0.4 | 36.1±0.4 | 39.9±0.3 | 36.9±0.8 | 26.6±0.5 | 27.1±0.5 |
| **DM + CHDDL** | **49.8**±0.5 | **39.0**±0.3 | **40.9**±0.7 | **39.8**±1.0 | **26.9**±0.5 | **27.4**±0.4 |
| MTT | 50.7±0.8 | 23.2±1.3 | 45.7±0.8 | 38.9±0.8 | 20.3±1.4 | 22.5±0.7 |
| **MTT + CHDDL** | **57.9**±0.4 | **24.0**±1.0 | **46.6**±0.9 | **41.1**±0.7 | **20.5**±0.8 | **22.8**±1.0 |

10 and PACS datasets under 10 IPC setting on DC+CHDDL in Figure 3. Note that $D_{\text{syn}}^{\text{dom}}$ is used only during dataset condensation process to generate $D_{\text{syn}}$ and that all the results with CHDDL in Tables 1, 2, and 3 are obtained when the model is trained with only the dataset $D_{\text{syn}}$ that has $M$ datapoints i.e. without using $D_{\text{sym}}^{\text{dom}}$ at all.

**Cross-architecture generalization**    We assess the generalization capabilities of condensed synthetic data across different architecture frameworks. Following MTT (Cazenavette et al., 2022), we experiment on ConvNet, AlexNet, VGG11, and ResNet-18. Furthermore, we experiment on ViT-Tiny and ViT-Small, which prior methods did not experiment on. The cross-architecture experiments are conducted with condensed CIFAR-10 data under 10 IPC with ConvNet. As shown in Table 3, incorporating CHDDL shows superior generalization performance across methods and architectures compared to those without CHDDL, demonstrating the robustness across architecture.

## 5    DISCUSSION

**Single-domain and multi-domain dataset**    The need for Multi-Domain Dataset Condensation (MDDC) methods has been highlighted in introduction as a figure. We further extend the experiment on the same setting and show the results in Table 4. For a single-domain dataset setting, we isolate the target domain with an explicit domain label for condensing and evaluating. On the other hand, for a multi-domain dataset setting, the whole PACS dataset (i.e., all four domains) is utilized for the training. The evaluation was done on the same target domains for both single- and multi-domain dataset settings. Applying CHDDL consistently narrows this performance gap across all baseline methods. Notably, in some cases, the performance becomes on-par with the single-domain setting, demonstrating effectiveness of CHDDL in preserving domain-specific features within a heterogeneous dataset.

**Leave-one-domain-out evaluation**    In this section, we evaluate whether embedding domain information into each synthetic image improves generalization to unseen domains beyond the training set. For the experiment, we tested on the three domain generalization benchmarks and compared DM

Table 4: Experiment results of single- and multi-domain dataset settings. In a single-domain dataset setting, the target data is used during the condensation process, whereas in a multi-domain dataset setting, the whole PACS dataset is utilized. The evaluation is done in the target domain images for both settings. The value inside the parentheses denotes the difference between Multi-Domain with CHDDL and without CHDDL.

| Method (↓) / Target Domain (→) | | Photo | Art-Painting | Cartoon | Sketch |
|---|---|---|---|---|---|
| Single-Domain | DC | 50.6 | 29.6 | 53.7 | 43.8 |
| Multi-Domain | DC | 48.1 | 27.6 | 38.0 | 32.1 |
| | **DC + CHDDL** | 49.4 (+1.3) | 30.4 (+2.8) | 40.4 (+2.4) | 37.4 (+5.3) |
| Single-Domain | DM | 50.7 | 29.5 | 50.2 | 35.4 |
| Multi-Domain | DM | 46.8 | 21.3 | 35.4 | 22.6 |
| | **DM + CHDDL** | 47.4 (+0.6) | 29.3 (+8.0) | 37.1 (+1.7) | 30.9 (+8.3) |
| Single-Domain | MTT | 55.2 | 31.9 | 55.1 | 42.3 |
| Multi-Domain | MTT | 50.7 | 24.5 | 40.8 | 44.6 |
| | **MTT + CHDDL** | 52.1 (+1.4) | 26.9 (+2.4) | 50.5 (+9.7) | 50.9 (+6.3) |

Table 5: Leave-one-domain-out evaluation on PACS, VLCS, and Office-Home datasets with 1 IPC using DM and DM+CHDDL. Target domains are abbreviated as: PACS — (P)hoto, (A)rt-Painting, (C)artoon, (S)ketch; VLCS — Pascal (V)OC, (L)abelMe, (C)altech, (S)un; Office-Home — (A)rt, (C)lipart, (P)roduct, (R)eal-World.

| Dataset | PACS | | | | VLCS | | | | Office-Home | | | |
|---|---|---|---|---|---|---|---|---|---|---|---|---|
| Target Domain | P | A | C | S | V | L | C | S | A | C | P | R |
| DM | 29.9 | 18.9 | 20.6 | 22.5 | 24.6 | 33.9 | 21.8 | 33.0 | 3.3 | 6.3 | 7.0 | 5.8 |
| **DM+CHDDL** | **44.4** | **24.4** | **27.7** | **30.7** | **26.9** | **40.0** | **26.0** | **36.3** | **5.2** | **7.4** | **9.7** | **7.1** |

Table 6: Comparison of different pseudo-domain labeling strategies on the CIFAR-10, PACS, and VLCS datasets under 1 and 10 IPC. All results are the average of 10 runs and reported as mean ± standard deviation. FFT: frequency feature extraction; log-Var: log-variance of early features; Mean-Sort: ordering features by mean value; K-Means: clustering features with K-Means. Baselines include random pseudo-labels and actual domain labels.

| Pseudo Domain Labeling Method | | | | Method | | | | | |
|---|---|---|---|---|---|---|---|---|---|
| | | | | CIFAR-10 | | PACS | | VLCS | |
| FFT | log-Var | Mean-Sort | K-Means | 1 IPC | 10 IPC | 1 IPC | 10 IPC | 1 IPC | 10 IPC |
| ✓ | | ✓ | | **27.1**±0.3 | **49.8**±0.5 | **34.7**±1.1 | **50.9**±0.4 | 36.7±1.1 | **44.4**±0.3 |
| ✓ | | | ✓ | 26.6±0.4 | 49.7±0.3 | 32.3±0.6 | 49.7±0.7 | **36.8**±1.1 | 44.3±0.4 |
| | ✓ | ✓ | | 27.0±0.3 | **49.8**±0.2 | 33.6±0.8 | 49.7±0.4 | 34.0±0.8 | 44.1±0.4 |
| | ✓ | | ✓ | 26.5±0.4 | 49.4±0.3 | 33.5±0.6 | 48.8±0.6 | 35.3±1.1 | 44.0±0.4 |
| Random Pseudo Labels | | | | 25.3±0.5 | 48.1±0.7 | 31.7±1.4 | 48.0±1.2 | 31.1±1.8 | 42.4±0.7 |
| Actual Domain Labels | | | | - | - | 34.0±1.7 | 50.6±0.6 | 34.5±1.5 | 43.9±0.5 |

with DM+CHDDL under 1 IPC using explicit domain labels only to isolate the target domain, which is only used during evaluation and neglected during the condensation process. As can be seen from Table 5, employing CHDDL with DM performed better with a substantial gap. This validates that employing CHDDL substantially increases the generalization ability of the condensed data through embedding informative and non-overlapping domain information. These results showcase the possibility of using CHDDL even for domain adaptation and domain generalization, where the burden of gathering data is much more costly.

**Various pseudo-domain labeling**  To evaluate the effectiveness of our pseudo-domain labeling strategy, we further experimented with log-variance (log-var) for extracting domain-specific features and K-Means clustering for clustering the extracted features to assign pseudo-domain labels. The feature for log-variance is extracted from the first and second layers of the three-depth ConvNet. Additionally, we compare the results with the random pseudo-domain labeling and actual domain labels for the available datasets, PACS and VLCS. The random pseudo-domain labeling is done by assigning a pseudo-domain label for each synthetic image not pixel-wise as done in CHDDL. The experiments are conducted using DM+CHDDL

across three datasets under 1 and 10 IPC and the results are shown in Table 6. Across all datasets and IPC configurations, FFT-based feature extraction consistently outperforms log-variance, regardless of the clustering strategy applied. Notably, the combination of FFT and Mean-Sort achieves the highest performance and even surpasses the use of actual domain labels.

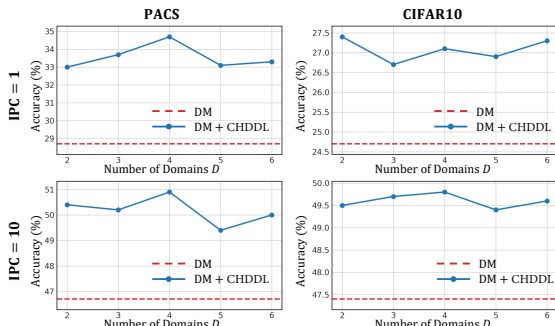

In contrast, random pseudo-domain labeling yields the lowest performance among all variants, though it still performs better than DM without CHDDL, highlighting the value of incorporating even weak domain information. Notably, our FFT-based pseudo-labeling outperforms using ground-truth domain labels. This suggests that the ground-truth labels (e.g., 'Art-Painting,' 'Sketch') might be too coarse for guiding condensation. Our method, by sorting images based on low-frequency statistics, creates more visually coherent groupings that are better aligned with the stylistic features relevant for condensation, even if they don't perfectly match human-defined categories.

Figure 4: Experiment with a varying number of domains $N_D$ on CIFAR-10 and PACS dataset under 1 and 10 IPC with DM and DM+CHDDL.

**Effect of the Number of Pseudo Domains** We analyze the impact of the number of pseudo domains $D$ on the performance of DM+CHDDL across CIFAR-10 and PACS under both 1 and 10 IPC. As shown in Figure 4, the dashed red line indicates the performance of the baseline DM method, while the solid blue curve shows the performance of DM+CHDDL as $D$ varies. In all settings, DM+CHDDL consistently outperforms DM, demonstrating the effectiveness of incorporating domain-specific information during condensation. Notably, for PACS, which has four explicit annotated domains, the best performance is observed when $D = 4$ in both IPC settings, suggesting that our pseudo-labeling scheme effectively captures the true underlying domain structure. On the other hand, the optimal number of pseudo domains in CIFAR-10 varies across settings, indicating that the best partitioning may depend on the nature of the dataset and the number of images per class, however, we emphasize that even with random number of domains $D$, we achieve better performance than basline methods (i.e. w/o CHDDL).

## 6 CONCLUSION

In this work, we addressed a critical yet overlooked challenge in dataset condensation: the performance degradation on heterogeneous, multi-domain datasets. We first formally defined this problem as the Multi-Domain Dataset Condensation (MDDC) task, establishing a new direction for creating robust and practical condensed datasets. As the first effective solution to this task, we proposed Condensing Heterogeneous Datasets without Domain Labels (CHDDL), a novel framework that embeds domain diversity into synthetic images. By unifying an unsupervised FFT-based pseudo-labeling scheme with learnable spatial masks, CHDDL successfully enhances the richness of synthetic data while operating without explicit domain labels or an increased images per class budget. Extensive experiments across various datasets, architectures, and base methods confirmed that CHDDL consistently improves both in-domain performance and out-of-domain generalization.

**Limitation** While CHDDL establishes a strong foundation, it introduces a moderate overhead in terms of condensation time and memory. As condensation is a one-time, offline process, this overhead is generally acceptable, but future work could explore more efficient implementations. Furthermore, extending CHDDL to dense prediction tasks like semantic segmentation remains a non-trivial challenge due to its current design for image-level classification. We believe that tackling these limitations presents exciting avenues for future research in creating truly universal and efficient condensed datasets.

ETHICS STATEMENT

This research aims to improve the computational and storage efficiency of training deep learning models. By reducing the resources required for training, our work can contribute positively to lowering the energy consumption and carbon footprint associated with large-scale machine learning. We used publicly available, standard benchmark datasets for all experiments. We do not foresee any direct negative ethical implications or societal consequences resulting from this work.

REPRODUCIBILITY STATEMENT

To ensure the reproducibility of our results, we have provided detailed descriptions of our methodology, experimental setup, and evaluation protocols in the main manuscript. All hyperparameters for our proposed method and for the baseline methods are detailed in the Appendix. The source code for our experiments are available in the supplementary zip file.

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

# A  APPENDIX

## A.1  DETAILS REGARDING $\Theta$ AND $\Theta'$

We clarify the roles of two sets of model parameters in our framework: $\Theta$ and $\Theta'$. Both parameter sets correspond to models with the same architecture but serve different purposes and operate on different types of data batches.

The need for $\Theta'$ arises from the fact that domain-aware loss must be computed over batches grouped by pseudo-domain labels (e.g., derived via FFT-based clustering), which differs from the class-wise batches typically used in condensation methods. Moreover, in methods such as DC and MTT, where $\Theta$ is either actively updated or pretrained for a specific matching loss, reusing the same parameter set for domain-aware supervision is unsuitable. Thus, $\Theta'$ is introduced to decouple domain-specific learning from class-based learning during the condensation process.

- **DC (Zhao et al., 2021)**: $\Theta$ is randomly initialized and updated through bi-level optimization using class-wise batches. Because $\Theta$ is trained throughout the condensation process, a separate parameter set $\Theta'$ is introduced and trained on domain-grouped batches to compute the domain-aware loss independently.
- **DM (Zhao & Bilen, 2023)**: $\Theta$ is randomly initialized but remains fixed throughout the condensation process. Since the parameters are not updated, the same $\Theta$ can be reused to compute the domain-aware loss, and an explicit $\Theta'$ is not required, even though domain-grouped batches are still used for the loss computation.
- **MTT (Cazenavette et al., 2022)**: $\Theta$ is pretrained on real data and used to guide condensation via stored training trajectories. To preserve this role, a separate parameter set $\Theta'$ is trained independently using **pseudo-domain labels** on real data prior to condensation, in a manner similar to the pretraining of $\Theta$.

Across all methods incorporating domain-aware learning, batches used with $\Theta'$ are consistently organized by pseudo-domain labels. Whether a distinct $\Theta'$ is needed depends on whether $\Theta$ is trained, before or during the condensation process.

Table A: Experiment results of SRe2L and SRe2L+CHDDL on PACS dataset under 10 and 50 IPCs.

| Dataset ($\rightarrow$) | PACS | |
|---|---|---|
| Img/Cls ($\rightarrow$) | 10 | 50 |
| Method ($\downarrow$) | | |
| SRe2L (Yin et al., 2023) | 21.3 | 22.3 |
| SRe2L + CHDDL | **24.2** | **28.3** |

## A.2  PSEUDOCODE

Algorithm 1 provides a detailed, step-by-step description of the CHDDL training procedure to facilitate reproducibility. The algorithm highlights the sequential nature of the class-guided and domain-guided update steps within each condensation iteration. Line 3 corresponds to the unsupervised labeling process detailed in Section 3.1. The main loop (Lines 5-16) describes the core optimization process, where the class-guided update (Lines 7-9) isolates the optimization of class-discriminative features, while the domain-guided update (Lines 12-14) injects domain diversity by optimizing both the synthetic images and the learnable masks. This two-step process directly implements the training objective defined in Section 3.3.

---

**Algorithm 1** The CHDDL Training Procedure

---

1: **Input:** Real dataset $\mathcal{D}_{real}$, images per class (IPC), num pseudo-domains $D$, loss weight $\lambda$.
2: **Output:** Condensed dataset $\mathcal{D}_{syn}$.
3: **Initialize:** Synthetic dataset $\mathcal{D}_{syn}$, learnable spatial masks $\{Z_m^d\}_{m,d}$.
4: Pre-compute pseudo-domain labels $\{d_n\}_{n=1}^N$ for $\mathcal{D}_{real}$ using the FFT-based method (Eq. 2).
5: **for** each condensation iteration **do**
6:     // === 1. Class-Guided Update Step ===
7:     **for** each class $c = 1, \ldots, C$ **do**
8:         Update $\mathcal{D}_{syn}$ by minimizing $\mathcal{L}_{cls}$ using batches corresponding to class $c$.
9:     **end for**
10:    // === 2. Domain-Guided Update Step ===
11:    **for** each pseudo-domain $d = 0, \ldots, D - 1$ **do**
12:       Generate domain views $\{\tilde{x}_m^d\}$ from the current $\mathcal{D}_{syn}$ and $\{Z_m^d\}$.
13:       Update both $\mathcal{D}_{syn}$ and $\{Z_m^d\}$ by minimizing $\mathcal{L}_{dom}$ using batches for domain $d$.
14:    **end for**
15: **end for**
16: **Return** $\mathcal{D}_{syn}$ (the masks $\{Z_m^d\}$ are discarded). =0

---

## A.3 APPLICATION TO RECENT DC METHOD

We test the compatibility of CHDDL with decoupling-based condensation methods, we integrate CHDDL into SRe2L (Yin et al., 2023), which reconstructs synthetic images by decoding cluster centroids in feature space. As shown in Table A, CHDDL consistently improves performance over vanilla SRe2L. Specifically, at 10 and 50 IPC on PACS, SRe2L achieves 21.3% and 22.3% accuracy, while SRe2L+CHDDL improves these to 24.2% and 28.3%, respectively. This highlights that domain-aware supervision complements decoupled condensation even in pipelines with non-parametric clustering and pretrained decoders. These results reinforce CHDDL's general applicability across diverse condensation paradigms.

## A.4 INITIALIZING SYNTHETIC DATA WITH REAL IMAGES

Table B: Performance comparison on CIFAR-10 and PACS datasets under 1 and 10 Image Per Class (IPC) settings. The experiment was done with **real** initializing the synthetic data following prior methods.

| Dataset | CIFAR-10 | | PACS | |
|---|---|---|---|---|
| Img/Cls | 1 | 10 | 1 | 10 |
| Ratio (%) | 0.02 | 0.2 | 0.08 | 0.8 |
| DC | $28.2_{\pm 0.6}$ | $44.7_{\pm 0.5}$ | $35.9_{\pm 1.1}$ | $47.7_{\pm 1.1}$ |
| DC + CHDDL | $\mathbf{29.0}_{\pm 0.4}$ | $\mathbf{45.2}_{\pm 0.3}$ | $\mathbf{37.8}_{\pm 0.7}$ | $\mathbf{48.7}_{\pm 0.2}$ |
| DM | $25.7_{\pm 0.6}$ | $49.1_{\pm 0.2}$ | $32.0_{\pm 1.9}$ | $50.0_{\pm 0.9}$ |
| DM + CHDDL | $\mathbf{26.8}_{\pm 0.3}$ | $\mathbf{50.0}_{\pm 0.3}$ | $\mathbf{32.7}_{\pm 1.2}$ | $\mathbf{50.9}_{\pm 0.6}$ |
| MTT | $45.4_{\pm 0.2}$ | $65.3_{\pm 0.4}$ | $44.3_{\pm 1.8}$ | $51.4_{\pm 1.2}$ |
| MTT + CHDDL | $\mathbf{45.6}_{\pm 0.3}$ | $\mathbf{65.5}_{\pm 0.2}$ | $\mathbf{44.4}_{\pm 1.6}$ | $\mathbf{53.5}_{\pm 1.3}$ |

In the main manuscript, all experiments initialize synthetic data using **Gaussian noise**, which better aligns with the privacy-preserving goals of dataset condensation. However, to demonstrate that our proposed method works even under alternative initializations, we conduct additional experiments where synthetic data is initialized with real images, selecting a random image from the corresponding class in the real dataset, following prior works. These results, shown in Table B, represent averages over 10 runs, consistent with our main evaluation protocol.

While all methods benefit from real initialization, as expected due to the additional structure provided at the start, the performance gains from CHDDL persist, underscoring its robustness. Notably, the relative improvement from CHDDL remains more pronounced in multi-domain settings like PACS, where domain shift presents a bigger challenge. In contrast, single-domain datasets such

as CIFAR-10 exhibit smaller domain-induced variability, which partially reduces the benefits of CHDDL when real images are used as initialization.

## A.5 HYPERPARAMETER SWEEP

### A.5.1 DOMAIN MASK INITIALIZATION $\mathbf{z}_m$

Table C: Effect of varying the domain mask initialization on CIFAR-10 and PACS datasets under IPC 1 and IPC 10. All results are the average of 10 runs and reported as mean $\pm$ standard deviation. The gray background setting is the setting equal to the results in the main manuscript. The highest is **bolded** and the second highest is underlined.

| Dataset ($\rightarrow$) | | CIFAR-10 | | PACS | |
|---|---|---|---|---|---|
| Img/Cls ($\rightarrow$) | | 1 | 10 | 1 | 10 |
| Ratio (%) ($\rightarrow$) | | 0.02 | 0.2 | 0.08 | 0.8 |
| Method ($\downarrow$) | Initial Value ($\downarrow$) | | | | |
| DC + CHDDL | 0.1 | $28.5_{\pm0.3}$ | $\underline{45.3_{\pm0.4}}$ | $38.5_{\pm0.9}$ | $\underline{48.8_{\pm0.7}}$ |
| | 0.05 | $28.8_{\pm0.3}$ | $45.2_{\pm0.4}$ | $38.8_{\pm0.9}$ | $47.9_{\pm0.7}$ |
| | 0.01 | $\mathbf{29.0_{\pm0.5}}$ | $\mathbf{45.4_{\pm0.3}}$ | $\underline{38.8_{\pm0.7}}$ | $48.3_{\pm0.5}$ |
| | 0.005 | $28.7_{\pm0.5}$ | $45.2_{\pm0.2}$ | $38.6_{\pm0.4}$ | $\mathbf{49.0_{\pm0.7}}$ |
| | 0.001 | $28.7_{\pm0.6}$ | $45.1_{\pm0.4}$ | $\mathbf{39.4_{\pm0.6}}$ | $47.9_{\pm0.6}$ |
| DM + CHDDL | 0.1 | $\mathbf{27.2_{\pm0.3}}$ | $49.7_{\pm0.3}$ | $34.2_{\pm1.6}$ | $50.1_{\pm0.5}$ |
| | 0.05 | $26.6_{\pm0.3}$ | $\mathbf{49.9_{\pm0.4}}$ | $\underline{34.2_{\pm0.5}}$ | $49.7_{\pm0.6}$ |
| | 0.01 | $\underline{27.1_{\pm0.3}}$ | $49.8_{\pm0.5}$ | $\mathbf{34.7_{\pm0.8}}$ | $\mathbf{50.9_{\pm0.4}}$ |
| | 0.005 | $26.3_{\pm0.3}$ | $48.9_{\pm0.2}$ | $34.2_{\pm1.0}$ | $49.2_{\pm0.7}$ |
| | 0.001 | $26.2_{\pm0.4}$ | $48.3_{\pm0.2}$ | $33.8_{\pm1.2}$ | $\underline{50.8_{\pm0.4}}$ |
| MTT + CHDDL | 0.1 | $\underline{46.4_{\pm0.4}}$ | - | $43.0_{\pm1.2}$ | - |
| | 0.05 | $\underline{41.1_{\pm0.6}}$ | - | $43.1_{\pm1.0}$ | - |
| | 0.01 | $\mathbf{46.8_{\pm0.4}}$ | $\mathbf{57.9_{\pm0.4}}$ | $\mathbf{46.6_{\pm1.3}}$ | $\mathbf{50.6_{\pm0.6}}$ |
| | 0.005 | $41.2_{\pm0.8}$ | - | $43.3_{\pm1.2}$ | - |
| | 0.001 | $41.9_{\pm0.5}$ | - | $\underline{40.7_{\pm1.2}}$ | - |

The domain mask initialization value ($z_m$) controls the initial scale of the softmax-normalized spatial masks applied in the Condensing Heterogeneous Datasets without Domain Labels (CHDDL). A smaller $z_m$ leads to nearly uniform domain weights at the beginning of training, allowing all domain masks to contribute equally. In contrast, a larger $z_m$ produces more confident, peaked softmax outputs early on, encouraging the model to assign higher importance to specific domains from the start. To understand the effect of this initialization, we conduct a sweep across a range of $z_m$ values and showcase the result in Table C.

We find that the setting used in the main manuscript ($z_m = 0.01$, gray-highlighted) consistently results in first- or second-best performance across all methods and datasets. Crucially, even under different initializations, methods with CHDDL uniformly outperform their respective baselines without CHDDL, indicating strong robustness. While MTT + CHDDL shows slightly more variation across $z_m$ values compared to other methods, it still maintains a clear performance margin over MTT without CHDDL. Due to observed instability at 1 IPC, we omit MTT + CHDDL results for IPC 10 in this ablation.

Overall, these results confirm that $z_m = 0.01$ is a reliable and effective choice, and that CHDDL consistently enhances performance across settings.

### A.5.2 DOMAIN EMBEDDING WEIGHT $\lambda$

We study the effect of varying the domain embedding weight $\lambda$, which balances the class loss and domain-aware loss in CHDDL. A smaller $\lambda$ reduces the influence of domain-specific learning, while a larger value encourages the model to attend more strongly to domain variations during condensation.

Table D: Effect of varying the embedding weight on CIFAR-10 and PACS datasets under IPC 1 and IPC 10. All results are the average of 10 runs and reported as mean $\pm$ standard deviation. The gray background setting is the setting equal to the results in the main manuscript. The highest is **bolded** and the second highest is underlined

| Dataset ($\rightarrow$) | | CIFAR-10 | | PACS | |
|---|---|---|---|---|---|
| Img/Cls ($\rightarrow$) | | 1 | 10 | 1 | 10 |
| Ratio (%) ($\rightarrow$) | | 0.02 | 0.2 | 0.08 | 0.8 |
| Method ($\downarrow$) | $\lambda$ ($\downarrow$) | | | | |
| DC + CHDDL | 1.0 | $27.9_{\pm 0.6}$ | $44.5_{\pm 0.5}$ | $37.1_{\pm 1.1}$ | $47.5_{\pm 0.6}$ |
| | 0.5 | $28.3_{\pm 0.7}$ | $45.0_{\pm 0.6}$ | $37.5_{\pm 1.0}$ | $47.3_{\pm 0.7}$ |
| | 0.1 | $\mathbf{29.0}_{\pm 0.5}$ | $45.4_{\pm 0.3}$ | $38.8_{\pm 0.7}$ | $\mathbf{48.3}_{\pm 0.5}$ |
| | 0.05 | $\mathbf{29.0}_{\pm 0.2}$ | $44.9_{\pm 0.3}$ | $\mathbf{38.9}_{\pm 0.7}$ | $47.7_{\pm 0.3}$ |
| | 0.01 | $28.9_{\pm 0.6}$ | $\mathbf{45.6}_{\pm 0.2}$ | $38.8_{\pm 0.4}$ | $48.0_{\pm 0.5}$ |
| | 0.005 | $28.8_{\pm 0.4}$ | $45.0_{\pm 0.2}$ | $38.3_{\pm 0.5}$ | $47.7_{\pm 0.5}$ |
| | 0.001 | $28.9_{\pm 0.5}$ | $\underline{45.7}_{\pm 0.3}$ | $38.6_{\pm 0.4}$ | $48.0_{\pm 0.4}$ |
| DM + CHDDL | 1.0 | $25.9_{\pm 0.5}$ | $48.5_{\pm 0.5}$ | $30.5_{\pm 1.3}$ | $46.9_{\pm 0.9}$ |
| | 0.5 | $26.0_{\pm 0.6}$ | $49.1_{\pm 0.6}$ | $31.2_{\pm 1.4}$ | $49.0_{\pm 1.0}$ |
| | 0.1 | $27.1_{\pm 0.3}$ | $\mathbf{49.8}_{\pm 0.5}$ | $\mathbf{34.7}_{\pm 0.8}$ | $\mathbf{50.9}_{\pm 0.4}$ |
| | 0.05 | $\mathbf{27.2}_{\pm 0.5}$ | $48.9_{\pm 0.4}$ | $34.3_{\pm 1.3}$ | $49.3_{\pm 0.4}$ |
| | 0.01 | $26.5_{\pm 0.2}$ | $49.0_{\pm 0.4}$ | $34.3_{\pm 0.6}$ | $50.4_{\pm 0.7}$ |
| | 0.005 | $26.7_{\pm 0.6}$ | $48.9_{\pm 0.3}$ | $34.3_{\pm 0.7}$ | $50.0_{\pm 0.6}$ |
| | 0.001 | $26.8_{\pm 0.4}$ | $\underline{49.5}_{\pm 0.4}$ | $33.3_{\pm 0.7}$ | $50.5_{\pm 0.8}$ |
| MTT + CHDDL | 0.1 | $46.1_{\pm 1.3}$ | - | $43.3_{\pm 0.8}$ | - |
| | 0.05 | $46.5_{\pm 0.8}$ | - | $43.9_{\pm 0.5}$ | - |
| | 0.01 | $\mathbf{46.8}_{\pm 0.4}$ | $\mathbf{57.9}_{\pm 0.4}$ | $\mathbf{46.6}_{\pm 0.9}$ | $\mathbf{50.6}_{\pm 0.6}$ |
| | 0.005 | $46.0_{\pm 0.5}$ | - | $46.0_{\pm 0.7}$ | - |
| | 0.001 | $\underline{46.7}_{\pm 0.8}$ | - | $\underline{46.5}_{\pm 1.0}$ | - |

As shown in Table D, performance remains strong across a wide range of $\lambda$ values, showing that the method is not overly sensitive to this hyperparameter. The setting used in the main manuscript ($\lambda = 0.1$, gray-highlighted) consistently achieves the best or second-best performance across datasets and methods. This confirms that $\lambda = 0.1$ is a reliable default, and that CHDDL provides robust improvements without requiring precise tuning.

We do not explore values of $\lambda$ greater than 0.1, as assigning excessive weight to domain supervision risks overshadowing class-discriminative learning. As emphasized in the main manuscript, CHDDL is designed to enrich class information with domain cues, not to compete with it.

As with the previous sweep, we omit MTT + CHDDL results for IPC 10 due to instability observed under 1 IPC setting.

### A.5.3    TEMPERATURE $\tau$

We ablate the softmax temperature $\tau$ in CHDDL, which controls the sharpness of domain assignment. A lower $\tau$ (e.g., 0.1) enforces peaked domain masks, while higher values (e.g., 1 or 5) blend domain cues more evenly.

As demonstrated in Table E, experiments with $\tau = 1$ and $\tau = 5$ show similar results, whereas the more discriminative setting $\tau = 0.1$ yields a vivid improvement in most configurations. As with the previous sweep, we omit MTT + CHDDL results for IPC 10 due to instability observed under 1 IPC setting.

### A.6    ADDITIONAL DATASET FOR MULTI-DOMAIN SETTING

To evaluate the scalability of our approach on a larger multi-domain dataset, we conduct experiments on DomainNet (Peng et al., 2019), a benchmark dataset comprising 345 classes across six distinct

Table E: Effect of varying the temperature $\tau$ on CIFAR-10 and PACS datasets under IPC 1 and IPC 10. All results are the average of 10 runs and reported as mean $\pm$ standard deviation. The gray background setting is the setting equal to the results in the main manuscript. The highest is **bolded** and the second highest is underlined

| Dataset ($\rightarrow$) | | CIFAR-10 | | PACS | |
|---|---|---|---|---|---|
| Img/Cls ($\rightarrow$) | | 1 | 10 | 1 | 10 |
| Ratio (%) ($\rightarrow$) | | 0.02 | 0.2 | 0.08 | 0.8 |
| Method ($\downarrow$) | $\tau$ ($\downarrow$) | | | | |
| DC + CHDDL | 0.1 | $\mathbf{29.0}_{\pm 0.5}$ | $\mathbf{45.4}_{\pm 0.3}$ | $\mathbf{38.8}_{\pm 0.7}$ | $48.3_{\pm 0.5}$ |
| | 1 | $28.5_{\pm 0.4}$ | $\underline{45.3}_{\pm 0.3}$ | $\mathbf{38.8}_{\pm 0.7}$ | $\mathbf{48.7}_{\pm 0.5}$ |
| | 5 | $\underline{28.7}_{\pm 0.4}$ | $45.1_{\pm 0.3}$ | $\mathbf{38.8}_{\pm 0.5}$ | $\mathbf{48.7}_{\pm 0.5}$ |
| DM + CHDDL | 0.1 | $\mathbf{27.1}_{\pm 0.3}$ | $\mathbf{49.8}_{\pm 0.5}$ | $\mathbf{34.7}_{\pm 1.1}$ | $\mathbf{50.9}_{\pm 0.4}$ |
| | 1 | $26.9_{\pm 0.5}$ | $49.1_{\pm 0.3}$ | $33.1_{\pm 1.0}$ | $49.8_{\pm 0.5}$ |
| | 5 | $26.8_{\pm 0.4}$ | $\underline{49.1}_{\pm 0.4}$ | $33.3_{\pm 1.0}$ | $\underline{50.4}_{\pm 0.5}$ |
| MTT + CHDDL | 0.1 | $\mathbf{46.8}_{\pm 0.4}$ | $\mathbf{57.9}_{\pm 0.4}$ | $46.6_{\pm 0.9}$ | $\mathbf{50.6}_{\pm 0.6}$ |
| | 1 | $\underline{42.4}_{\pm 0.6}$ | - | $46.9_{\pm 0.4}$ | - |
| | 5 | $\underline{42.0}_{\pm 0.7}$ | - | $\mathbf{48.5}_{\pm 0.8}$ | - |

Table F: Comparison of DomainNet under IPC 1. All results are the average of 3 runs and reported as mean $\pm$ standard deviation.

| Dataset | DomainNet Peng et al. (2019) |
|---|---|
| Img/Cls | 1 |
| Ratio (%) | 0.06 |
| DM | $3.44_{\pm 0.03}$ |
| DM + CHDDL | $\mathbf{3.52}_{\pm 0.03}$ |

domains: Clipart, Infograph, Painting, Quickdraw, Real, and Sketch. The total dataset contains approximately 586,575 images, with the number of samples per domain ranging from 48,837 to 175,327, making it one of the largest and most diverse domain generalization datasets.

Due to the high computational demand of such a large-scale dataset, we perform the evaluation under the 1 Image Per Class (IPC) setting, which corresponds to a 0.06% data ratio. The results are reported in Table F. While the overall performance is lower, owing to the dataset's complexity and extreme data compression, the incorporation of CHDDL still provides a measurable improvement over the baseline DM method, further demonstrating the robustness and scalability of our proposed approach.

## A.7 COMPUTATIONAL COST

We report the GPU memory usage and per-iteration training time for with and without our method, CHDDL. All experiments were conducted using an *NVIDIA RTX A6000* GPU and an *Intel Xeon Gold 6442Y* CPU. The reported training time is the average duration of a single training loop measured over 10 iterations, taken after 10 warm-up iterations. Peak GPU memory consumption is measured during the same window using PyTorch's memory profiling utilities (`torch.cuda.max_memory_allocated()`).

As shown in Table G and Table H, and also noted in the limitation, incorporating CHDDL introduces an overhead. As we introduce the domain masks per image, GPU memory usage increases with images per class (IPC) and the number of dataset classes. However, the training time doubles only in the low IPC and does not linearly grow with the IPC and the number of dataset classes.

For MTT, we observed a slightly different behavior. GPU memory usage and training time were unstable across repeated runs, with noticeable fluctuations. We attribute this instability to the over-

Table G: Results with and without CHDDL on the prior methods on the **single-domain** setting. "T.Image." denotes Tiny ImageNet dataset. The results are shown as peak GPU consumption - average training loop time.

| Dataset | CIFAR-10 | | | CIFAR-100 | | | T.Image. |
|---|---|---|---|---|---|---|---|
| Img/Cls | 1 | 10 | 50 | 1 | 10 | 50 | 1 |
| DC | 1GiB - 0.2s | 1GiB - 11.1s | 2GiB - 60.6s | 2GiB - 1.8s | 5GiB - 105s | - | - |
| **DC + CHDDL** | 1GiB - 0.4s | 2GiB - 13.4s | 9GiB - 90.9s | 2GiB - 2.4s | 18GiB - 119s | - | - |
| DM | 0.1GiB - 0.1s | 1GiB - 0.1s | 1GiB - 0.1s | 1GiB - 0.7s | 2GiB - 0.8s | 8GiB - 0.8s | 6GiB - 3.5s |
| **DM + CHDDL** | 0.1GiB - 0.1s | 1GiB - 0.2s | 4GiB - 0.3s | 1GiB - 1.2s | 7GiB - 1.4s | 36GiB - 2.5s | 10GiB - 5.4s |
| MTT | 1GiB - 2.2s | 5GiB - 1.3s | - | 5GiB - 2.4s | - | - | - |
| **MTT + CHDDL** | 1GiB - 3.3s | 9GiB - 4.7s | - | 9GiB - 4.5s | - | - | - |

Table H: Results with and without CHDDL on the prior methods on the **multi-domain** setting. The results are shown as peak GPU consumption - average training loop time.

| Dataset | PACS | | VLCS | | Office-Home | |
|---|---|---|---|---|---|---|
| Img/Cls | 1 | 10 | 1 | 10 | 1 | 10 |
| DC | 3GiB - 0.4s | 4GiB - 26.6s | 3GiB - 0.3s | 3GiB - 18.9s | 4GiB - 2.9s | - |
| **DC + CHDDL** | 3GiB - 0.7s | 5GiB - 31.4s | 3GiB - 0.5s | 4GiB - 22.9s | 5GiB - 3.5s | - |
| DM | 1GiB - 0.1s | 1GiB - 0.2s | 1GiB - 0.1s | 1GiB - 0.1s | 2GiB - 1s | 5GiB - 1.1s |
| **DM + CHDDL** | 1GiB - 0.2s | 2GiB - 0.2s | 1GiB - 0.2s | 2GiB - 0.3s | 2GiB - 1.2s | 10GiB - 1.9s |
| MTT | 1GiB - 2.5s | 13GiB - 2.3s | 1GiB - 2.8s | - | 12GiB - 3.0s | - |
| **MTT + CHDDL** | 3GiB - 4.0s | 25GiB - 5.1s | 2GiB - 4.1s | - | 24GiB - 9.5s | - |

Table I: Hyperparameters for CHDDL with MTT. "T.Image." denotes Tiny ImageNet dataset and "OH" denote Office Home dataset.

| Dataset | IPC | Synthetic Steps | Exper Epochs | Max Start Epochs | Learning Rate Image | Learning Rate | Starting Synthetic Step Size |
|---|---|---|---|---|---|---|---|
| CIFAR-10 | 1 | 50 | 2 | 2 | 100 | $10^{-7}$ | $10^{-2}$ |
| | 10 | 30 | 2 | 20 | $10^5$ | $10^{-6}$ | $10^{-2}$ |
| CIFAR-100 | 1 | 20 | 3 | 20 | $10^3$ | $10^{-5}$ | $10^{-2}$ |
| | 10 | 20 | 2 | 20 | $10^3$ | $10^{-5}$ | $10^{-2}$ |
| T.Image. | 1 | 10 | 2 | 10 | $10^4$ | $10^{-4}$ | $10^{-2}$ |
| PACS | 1 | 10 | 2 | 10 | $10^4$ | $10^{-5}$ | $10^{-2}$ |
| | 10 | 20 | 2 | 40 | $10^4$ | $10^{-6}$ | $10^{-2}$ |
| VLCS | 1 | 10 | 2 | 10 | $10^4$ | $10^{-6}$ | $10^{-2}$ |
| | 10 | 20 | 2 | 40 | $10^4$ | $10^{-6}$ | $10^{-2}$ |
| OH | 1 | 10 | 2 | 10 | $10^4$ | $10^{-4}$ | $10^{-2}$ |

head of loading and processing trajectory data within each training loop, which is unique to the MTT framework. Due to this inconsistency, we report the highest observed GPU memory usage and training time across three repeated runs for each setting.

## A.8 HYPERPARAMETER FOR MTT

For DC and DM, we adopted the hyperparameters used in their respective original implementations. For multi-domain datasets, we followed the same configuration as used for Tiny ImageNet. In contrast, MTT required a separate hyperparameter search due to frequent occurrences of `NaN` losses

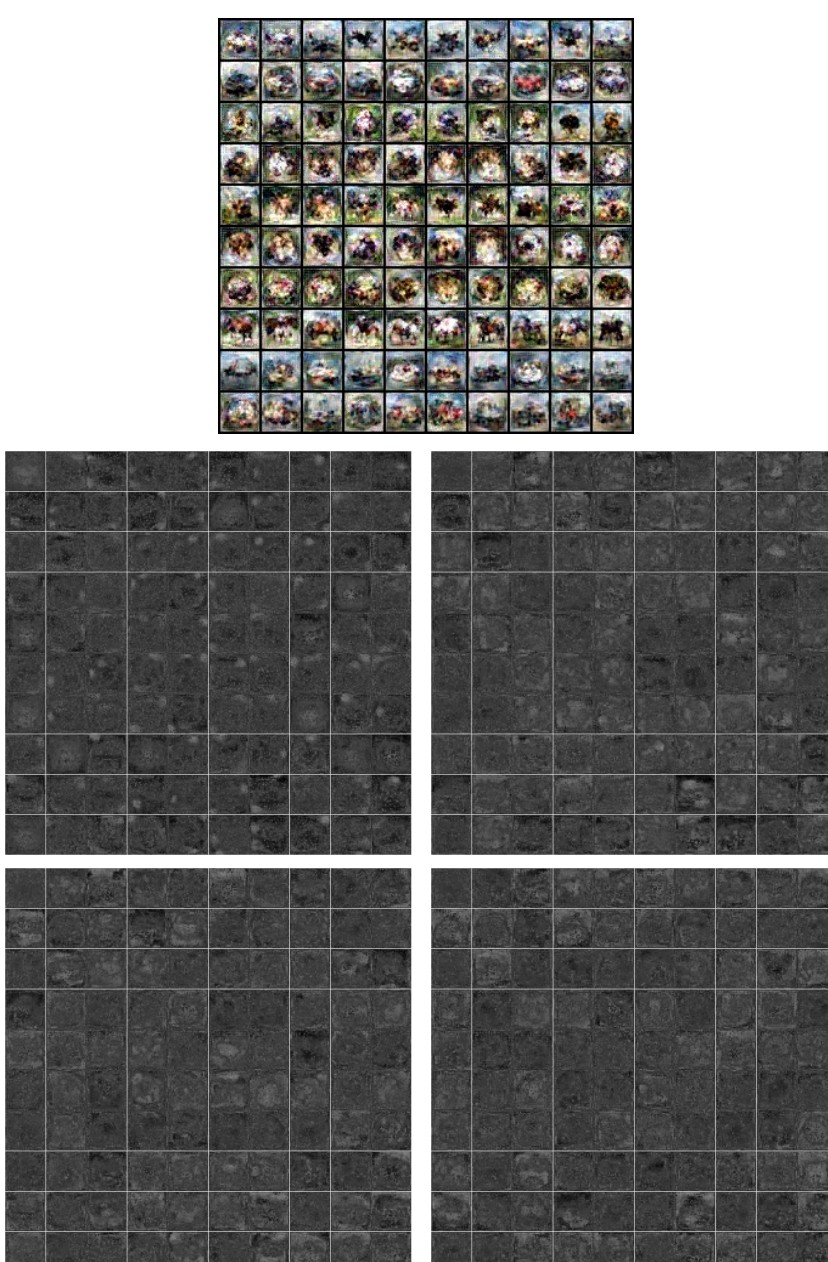

Figure A: Visualization of the final output and domain masks in CIFAR-10 under 10 IPC setting. The shown images are condensed with DC+CHDDL.

during training when combined with CHDDL and **Gaussian noise** initialization. Table I lists the hyperparameters used for MTT with CHDDL across all datasets.

We initially started with the settings reported in the original MTT paper (Cazenavette et al., 2022), and conducted minimal adjustments only when instability (e.g., NaN gradients or diverging loss) was observed. We constrained the search to a narrow range around the original values, preferring stability over aggressive tuning. It is important to note that these are not hyperparameters introduced by our method (CHDDL) but rather those that existed from the MTT pipeline.

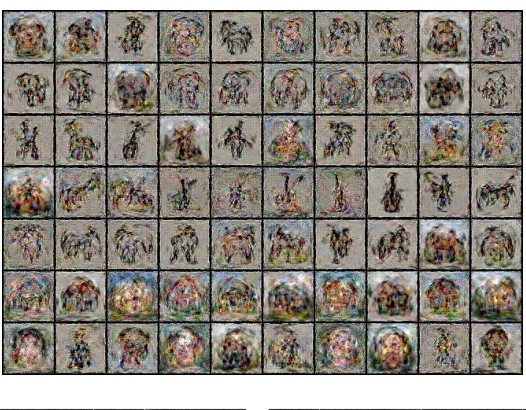

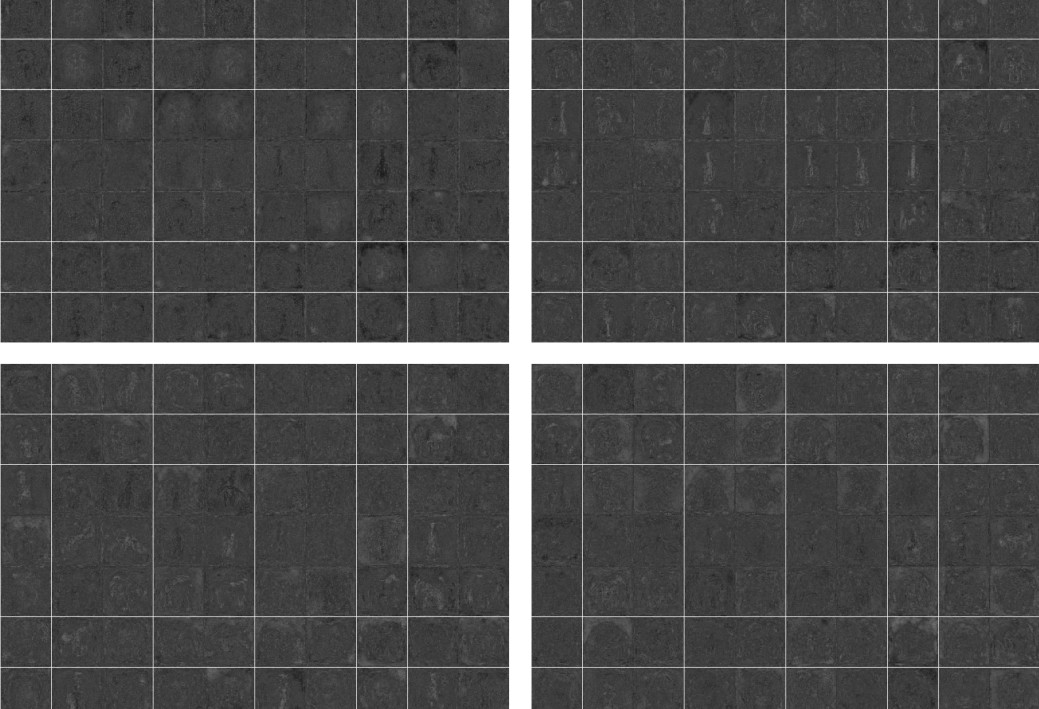

Figure B: Visualization of the final output and domain masks in PACS under 10 IPC setting. The shown images are condensed with DC+CHDDL.

## A.9 QUALITATIVE RESULTS

We provide additional qualitative examples of the condensed synthetic images generated with CHDDL in Figure A, B, C, and D.

All visualizations are obtained under IPC 10 using the CIFAR-10 and PACS datasets. We present results based on DC and DM baselines, and visualize the synthetic images and domain mask after the final condensation step.

## A.10 THE USE OF LARGE LANGUAGE MODELS (LLMS) NO LARGE LANGUAGE MODELS (LLMS) WERE USED IN THE DEVELOPMENT OF THE METHODOLOGY, THE EXECUTION OF THE EXPERIMENTS, OR THE ANALYSIS OF THE RESULTS PRESENTED IN THIS PAPER. WHILE WE LEVERAGED LLMS FOR MINOR WRITING REFINEMENTS.

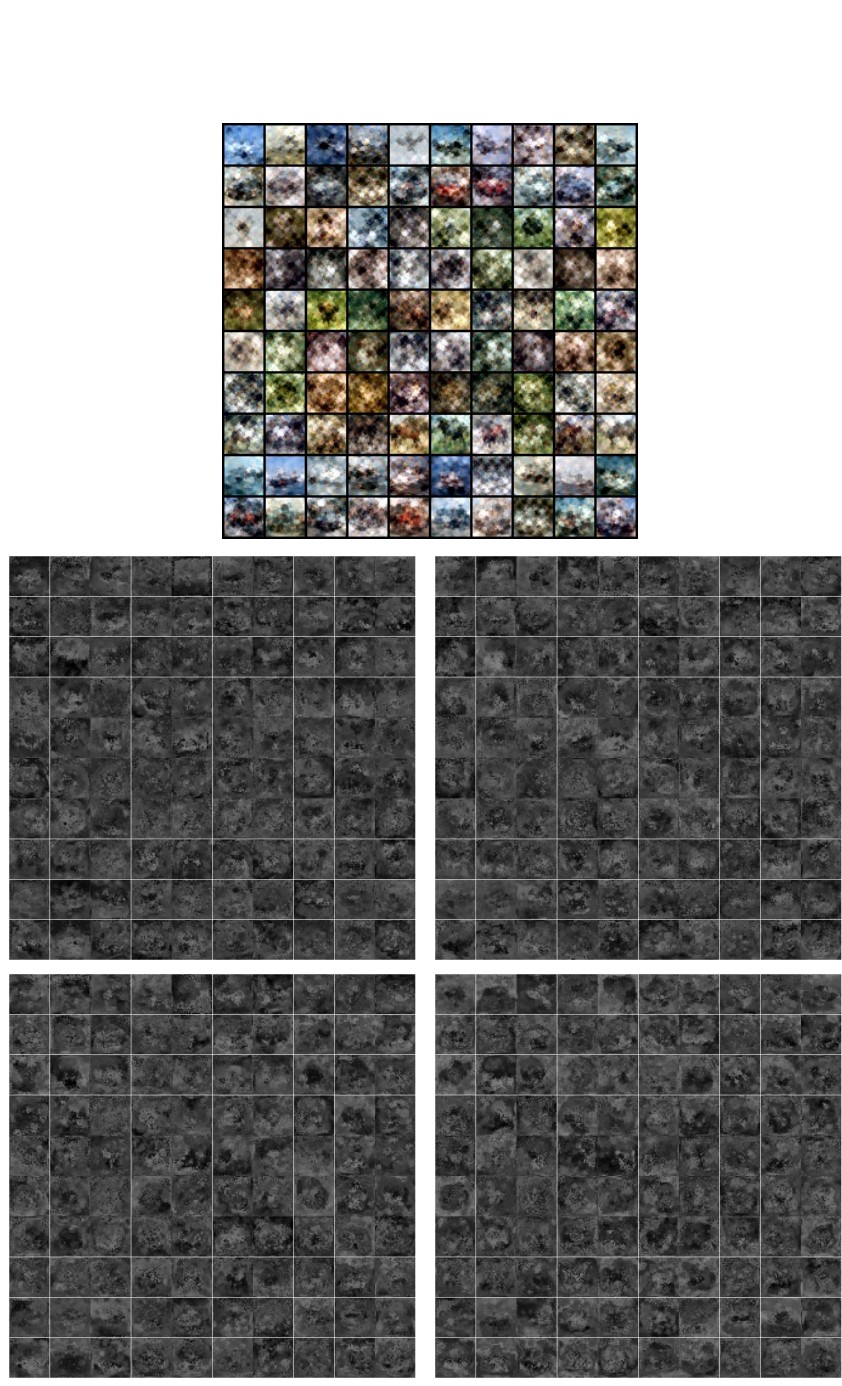

Figure C: Visualization of the final output and domain masks in CIFAR-10 under 10 IPC setting. The shown images are condensed with DM+CHDDL.

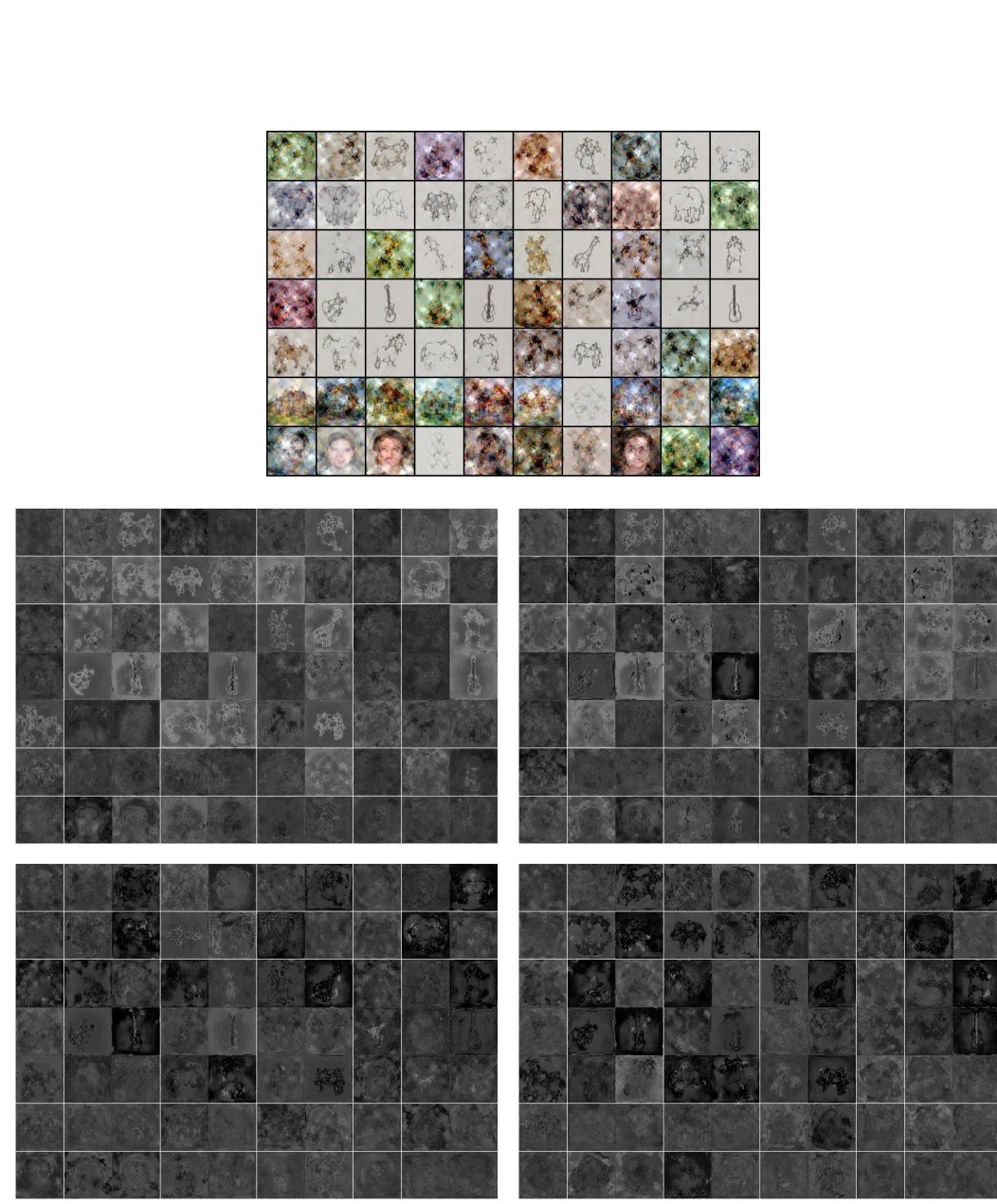

Figure D: Visualization of the final output and domain masks in PACS under 10 IPC setting. The shown images are condensed with DM+CHDDL.

