# OpenReview forum: "Condensing Heterogeneous Datasets without Domain Labels"
_ICLR.cc/2026/Conference — ICLR 2026 Conference Withdrawn Submission_

### Official Review · Reviewer_HSj2 · 2025-10-22

**Soundness:** 3
**Presentation:** 4
**Contribution:** 3
**Rating:** 6
**Confidence:** 4

**Summary:**

Existing Dataset Condensation (DC) methods often struggle with performance when applied to heterogeneous, multi-domain datasets. To address this issue, this paper introduces CHDDL, a method that enhances synthetic images with domain diversity without requiring explicit domain labels. The key innovations of CHDDL include an FFT-based pseudo-domain labeling scheme that identifies domain groups and the incorporation of rich domain knowledge into synthetic images through a domain-aware module. Experiments conducted on both single-domain and multi-domain benchmarks demonstrate that CHDDL consistently improves the performance of various DC baselines.

**Strengths:**

1. This paper first defines the Multi-Domain Dataset Condensation (MDDC) task, which aims to identify the performance degradation of conventional DC methods in heterogeneous settings.
2. The paper combines a simple yet effective frequency-based pseudo domain labeling method with a domain-aware module that encodes diversity information from multiple domains into synthetic images without requiring domain labels or additional IPC budget.
3. The paper presents well-designed experiments illustrating the method's effectiveness and general applicability across various datasets, domain settings, architectures, and base methods. Additionally, thorough ablation studies, hyperparameter sweeps, and computational analysis further demonstrate the method's robustness.

**Weaknesses:**

1. The related work section lacks comprehensive coverage. Since the pseudo domain labeling method is discussed in the Methods section, it is important to reference other pseudo domain labeling methods, besides the FFT-based approach, in the related work section. Currently, these alternatives only appear in the Discussion section. Including them in the related work would clarify why you chose a frequency-based approach over the other methods.
2. Some parts of the experiments do not provide a suitable comparison with existing methods. To my knowledge, previous approaches have primarily used real images for initialization, while this study has mainly focused on results that utilize Gaussian noise initialization. Furthermore, the discussion regarding real image initialization is only briefly addressed in the appendix.

**Questions:**

1. Why does CHDDL outperform other methods on single-domain datasets like CIFAR-10, without being influenced by external domain information?
2. In the context of frequency-based pseudo domain labeling, how does the cropping ratio β affect the model's performance, and what is the best method to determine the optimal β?
3. In Table B of the appendix, why does the model's performance improvement seem less significant compared to the Gaussian noise initialization when using real images to initialize synthetic data?
4. In Table E of the appendix, why does the model's performance almost remain stable when the temperature is varied? Additionally, how would the model's performance be affected if the temperature is less than 0.01?

---

### Official Review · Reviewer_8we9 · 2025-10-23

**Soundness:** 2
**Presentation:** 2
**Contribution:** 2
**Rating:** 2
**Confidence:** 5

**Summary:**

The paper defines Multi‑Domain Dataset Condensation (MDDC) and proposes CHDDL, a plug‑and‑play module that augments existing dataset condensation (DC) methods (DC/DM/MTT) with domain‑aware supervision without using domain labels or increasing images‑per‑class (IPC). Technically, CHDDL (i) derives pseudo‑domain labels from each real image using the mean low‑frequency amplitude of the FFT (crop ratio β), then sorts into D bins; (ii) learns per‑image, per‑pixel spatial masks (softmax temperature τ) to embed multiple pseudo‑domain “views” into each synthetic image; and (iii) optimizes a combined loss—the base condensation loss for class information and a domain loss computed on domain‑masked variants—while discarding the masks after condensation, so downstream training sees only the standard synthetic set.

Experiments on CIFAR‑10/100, Tiny‑ImageNet (single‑domain) and PACS, VLCS, Office‑Home (multi‑domain) show consistent but often modest gains when adding CHDDL to DC/DM/MTT at IPC∈{1,10,50}, plus cross‑architecture transfer (AlexNet/VGG/ResNet/ViT) from ConvNet‑distilled sets. The paper also includes ablations on the pseudo‑labeling strategy (FFT vs. alternatives), number of pseudo‑domains D, mask temperature τ, and training overheads; an appendix reports results on DomainNet (IPC=1) and integration with SRe2L.

**Strengths:**

Clear problem statement (MDDC) & practical constraints. The paper explicitly highlights that conventional DC collapses to dominant styles in heterogeneous datasets, motivates the need to encode domain variation without labels and without increasing IPC, and frames this as the MDDC task.

Method is simple, composable, and minimally invasive. CHDDL wraps around existing DC pipelines (DC/DM/MTT), adds a light domain‑aware branch during condensation, then discards masks—downstream training is unchanged. This design choice is appealing in practice.

Broad empirical coverage and consistent deltas. Results span six benchmarks with multiple base methods and IPC budgets; CHDDL almost always improves top‑1 accuracy in both single‑ and multi‑domain settings (e.g., PACS/VLCS/Office‑Home), with additional leave‑one‑domain‑out evaluations supporting improved out‑of‑domain generalization

Useful ablations and diagnostics. The paper probes pseudo‑labeling variants (FFT vs. log‑Var/K‑means), the number of pseudo‑domains D, temperature τ, initialization, and reports GPU/time overhead tables—rare but valuable for this literature.

Cross‑architecture transfer. Showing that ConvNet‑distilled sets help other architectures (AlexNet/VGG/ResNet/ViT) strengthens the external validity of the condensed data, even if absolute accuracies are still moderate.

**Weaknesses:**

Pseudo‑domain labeling is overly simplistic and potentially brittle. The single scalar (mean low‑frequency amplitude) used to rank/slice images into pseudo‑domains is a very coarse statistic that likely correlates with brightness/exposure/compression as much as “style” or “domain.” The paper does not analyze failure modes (e.g., illumination, JPEG quality, color cast) nor provide qualitative evidence that the FFT mean partitions align with coherent domain variation beyond a few visualizations.

Mask learning lacks regularization and interpretability guarantees. The per‑pixel softmax masks have no spatial smoothness or sparsity priors; the paper does not report whether masks become noisy/fragmentary or collapse. Because masks are discarded at the end, the method relies on the synthetic image itself to “bake in” diverse domain cues—yet no metric quantifies whether this “embedded diversity” persists.

Gains are often modest; absolute performance remains low in realistic settings.
On CIFAR‑100 (IPC=10), DM improves 29.2→30.0 (+0.8), and DC 24.8→25.8 (+1.0). On Tiny‑ImageNet (IPC=1), DM 3.7→4.2 (+0.5). These deltas, while consistent, are small against already low accuracies for aggressive condensation.
On DomainNet (IPC=1), the improvement is 3.44→3.52 (+0.08 absolute), which is statistically tiny and questions scalability to truly challenging, high‑diversity data.

Computational overhead undermines the “lightweight” narrative. For DM, Tiny‑ImageNet memory rises from 8 GiB to 36 GiB and loop time from 0.8 s to 2.5 s; for other settings, memory/time increases are often 2–4×. CHDDL is “one‑off” at condensation time, but the overhead is non‑trivial and may be prohibitive at larger resolutions/classes.

Choice of hyperparameters (D, β, τ, λ) lacks principled selection. The main runs fix D=4 across datasets (except a dedicated ablation), which conveniently matches PACS but is unlikely optimal for others; the paper offers empirical sweeps but no guidance or validation criterion for choosing D (or β/τ/λ) without tuning on test‑like signals.

Evaluation scope omits several stronger/modern baselines. The core comparisons are to DC (gradient matching), DM (distribution matching), and MTT (trajectory matching). The paper mentions recent generative approaches and decoupled methods, but the main text does not compare to them (SRe2L appears only in the appendix, and gains remain modest). A more competitive SOTA canvas is needed in 2025/26.

Method coupling and loss design remain under‑specified in the main text. The role of the two parameter sets Θ vs. Θ′ (when they share/freeze/are pretrained) is relegated to the appendix, which is important for reproducibility and may materially impact outcomes/overheads. This should be promoted to the main paper.

Potential confounds in the “CHDDL beats ground‑truth domain labels” claim. In Table 6, FFT‑based labels sometimes surpass actual domain labels, which is counterintuitive; the likely reason is label coarseness (e.g., ‘Art‑Painting’ mixes many substyles). But the experiment set‑up could also favor FFT ordering (e.g., different batching/statistics). A stronger test would compare against pseudo‑domains derived from self‑supervised features (e.g., CLIP/DINO style descriptors) rather than early ConvNet statistics or FFT means.

Limited resolution and architectural diversity for condensation. Most results operate at 32–64 px with shallow ConvNets as the distillation teacher—useful but far from modern training regimes. Cross‑architecture tables show small or inconsistent gains on ViTs, raising questions about how well CHDDL scales to higher‑capacity models.

Instability with MTT and practical tuning burden. The appendix notes NaNs and the need for bespoke hyperparameter search for MTT+CHDDL, which undercuts the “plug‑and‑play” claim in at least one important baseline.

**Questions:**

Why does FFT‑Mean sometimes beat ground‑truth domain labels? Beyond “coarseness,” can you show t‑SNE/UMAPs or intra‑/inter‑cluster style metrics that explain this surprising outcome in Table 6? Would CLIP/DINO‑based pseudo‑domains narrow or widen the gap?

How should practitioners choose D and β without labels? Provide an unsupervised model‑selection procedure (e.g., information criteria on FFT/feature clusters). Current fixed D=4 is ad‑hoc for non‑PACS datasets ?

Do masks become spatially coherent? Can you report statistics (e.g., number/size of connected components, TV norm) or show per‑class aggregates of masks to verify that the model learns meaningful spatial partitioning rather than diffuse blends?

Overheads at higher resolution? What happens at 128–224 px with standard backbones? Extrapolate memory/time as a function of resolution, IPC, and D, and discuss feasibility.

MTT instability. What failure modes caused NaNs with Gaussian init? Could gradient clipping, mixed‑precision tweaks, or smaller τ resolve this without extensive hyperparameter search?

Effect under severe domain imbalance. If one pseudo‑domain vastly dominates, does the per‑pixel softmax (with fixed τ) still prevent collapse? Any results where you synthetically skew domain frequencies?

Downstream fairness & bias. Since FFT‑based labels may correlate with illumination/skin tone/backgrounds, did you assess whether CHDDL amplifies or mitigates spurious correlations in the condensed set?

Ablation on λ with stronger backbones. The λ sweep shows robustness on small models; is the same true with ResNet‑50/ViT students where feature geometry differs?

---

### Official Review · Reviewer_Ra1M · 2025-10-26

**Soundness:** 2
**Presentation:** 1
**Contribution:** 2
**Rating:** 2
**Confidence:** 5

**Summary:**

This paper addresses the problem of dataset distillation in scenarios where data samples are collected from multiple domains. The authors argue that existing methods often collapse toward dominant domains, which limits performance on other domains. To overcome this issue, the paper proposes a masking-based module that transforms synthetic samples into multiple pseudo-domains. The resulting images are then supervised using real samples assigned to corresponding pseudo-domains, defined by the mean amplitude of their low-frequency components, and this supervision is used to update the original synthetic samples. Experimental results show that the proposed approach improves performance on both single-domain and multi-domain benchmarks.

**Strengths:**

- This paper is among the first to explore dataset condensation in a multi-domain setting.
- The proposed method introduces a novel way of grouping real samples based on frequency analysis and leveraging such group-wise information for supervising synthetic data.

**Weaknesses:**

- The paper does not clearly explain the rationale behind the design choices. While the method parameterizes each synthetic sample as a weighted sum of samples from different pseudo-domains, the underlying intuition or guiding principles are not sufficiently articulated.

- It is not entirely clear whether the method is specifically suited for multi-domain scenarios. The reported improvements are similar across single- and multi-domain settings, and the evidence that existing dataset condensation methods fail in multi-domain cases is not strongly established. While Fig. 1 shows performance degradation when condensing samples from multiple domains, this outcome is somewhat expected since mixing domains can naturally reduce single-domain accuracy. To strengthen the argument, it would be valuable to include results from training on the full real dataset in both single- and multi-domain settings as a reference.

- The experimental evaluation appears somewhat limited. Comparisons are made mainly against older baselines (over three years old), and in some cases the implementations differ from the originals (e.g., initializing images with noise rather than real samples in MTT and DM) without clear justification. Furthermore, while the authors mention computational constraints, stronger evaluations (such as 50 IPC on CIFAR-10/100) would be feasible with moderate resources (e.g., several 2080Ti GPUs), and could substantially strengthen the empirical validation.

- The presentation could be improved. Some claims are made without sufficient supporting evidence, such as the statement that synthetic images collapse toward dominant visual styles. Several notations are introduced without clear definitions (e.g., C in line 159, 'shifted' and `Crop_β` in Eq. (2), the use of superscript i across symbols, or operations like 'ranking' and the floor function in line 196). Certain formulations are also difficult to interpret (e.g., Eq. (2)), and parts of the exposition raise questions (e.g., the role of Table 1 in illustrating domain-specific structure in lines 319–321, the assignment of pseudo-domain labels to synthetic rather than real images in lines 430–431, or the claim that random labels incorporate weak domain information in lines 439-442). Finally, some statements appear inaccurate or require clarification, such as the description of IDM as addressing class imbalance (line 125).

**Questions:**

See weaknesses.

---

### Official Review · Reviewer_hSWG · 2025-11-03

**Soundness:** 3
**Presentation:** 3
**Contribution:** 3
**Rating:** 6
**Confidence:** 3

**Summary:**

The paper introduces the overlooked problem of dataset condensation on heterogeneous, unlabeled multi-domain data by introducing the multi-domain dataset condensation (MDDC) task. It proposes CHDDL, a framework that embeds domain diversity into synthetic images using FFT-based pseudo-labeling and learnable spatial masks. Extensive experiments show that CHDDL improves both in-domain and cross-domain performance across multiple benchmarks and condensation methods.

**Strengths:**

(1) This paper introduces the novel and practically relevant multi-domain dataset condensation problem, addressing a clear gap in existing dataset condensation research.

(2) The proposed framework, CHDDL, is a simple yet effective plug-and-play module that consistently enhances performance across diverse datasets and base methods without requiring domain labels.

**Weaknesses:**

See questions

**Questions:**

(1) How sensitive is the proposed method CHDDL to incorrect D or the true domain count is unknown?

(2) Could it be applied to large-scale datasets like ImageNet and its variants (ImageNetv2, ImageNet-sketch, ImageNet-adversarial, ImageNet-rendition) ? Will the FFT-based pseudo-labeling still be scaled efficiently?

---

### Note · Authors · 2025-11-13

I have read and agree with the venue's withdrawal policy on behalf of myself and my co-authors.